# Saprotrophic Wood Decay Ability and Plant Cell Wall Degrading Enzyme System of the White Rot Fungus *Crucibulum laeve*: Secretome, Metabolome and Genome Investigations

**DOI:** 10.3390/jof11010021

**Published:** 2024-12-31

**Authors:** Alexander V. Shabaev, Olga S. Savinova, Konstantin V. Moiseenko, Olga A. Glazunova, Tatyana V. Fedorova

**Affiliations:** A.N. Bach Institute of Biochemistry, Research Center of Biotechnology, Russian Academy of Sciences, Moscow 119071, Russia; a.shabaev1998@gmail.com (A.V.S.); savinova_os@rambler.ru (O.S.S.); mr.moiseenko@gmail.com (K.V.M.); olga.a.glas@gmail.com (O.A.G.)

**Keywords:** white rot fungi, *Crucibulum laeve* LE-BIN 1700, lignocellulose, secretome, exoproteome, metabolome, in silico sugar catabolism analysis

## Abstract

The basidiomycete *Crucibulum laeve* strain LE-BIN1700 (Agaricales, *Nidulariaceae*) is able to grow on agar media supplemented with individual components of lignocellulose such as lignin, cellulose, xylan, xyloglucan, arabinoxylan, starch and pectin, and also to effectively destroy and digest birch, alder and pine sawdust. *C. laeve* produces a unique repertoire of proteins for the saccharification of the plant biomass, including predominantly oxidative enzymes such as laccases (family AA1_1 CAZymes), GMC oxidoreductases (family AA3_2 CAZymes), FAD-oligosaccharide oxidase (family AA7 CAZymes) and lytic polysaccharide monooxygenases (family LPMO X325), as well as accompanying acetyl esterases and loosenine-like expansins. Metabolomic analysis revealed that, specifically, monosaccharides and carboxylic acids were the key low molecular metabolites in the *C. laeve* culture liquids in the experimental conditions. The proportion of monosaccharides and polyols in the total pool of identified compounds increased on the sawdust-containing media. Multiple copies of the family AA1_1, AA3_2, AA7 and LPMOs CAZyme genes, as well as eight genes encoding proteins of the YvrE superfamily (COG3386), which includes sugar lactone lactonases, were predicted in the *C. laeve* genome. According to metabolic pathway analysis, the litter saprotroph *C. laeve* can catabolize D-gluconic and D-galacturonic acids, and possibly other aldonic acids, which seems to confer certain ecological advantages.

## 1. Introduction

Lignocellulosic biomass (lignocellulose, LB) is the world’s largest carbon reservoir, and its major components (i.e., cellulose, hemicellulose and lignin) are among the most abundant biopolymers on Earth [1]. Lignocellulose is an extremely recalcitrant biopolymer that can often be degraded over many years or even decades. This is mainly due to its specific chemical composition and structure [2]. White rot Basidiomycetes (hereafter—white rot fungi, WRF) are the most efficient natural LB-decomposing organisms [3]. They include representatives of different ecological groups (e.g., phytoparasites, xylotrophs, humus and litter saprotrophs, mycorrhiza-forming species, etc.). Xylotrophic WRF, with a potent enzyme complex that allows complete mineralisation of all the lignocellulose components to carbon dioxide and water, play a leading role in LB degradation [4,5,6].

In recent decades, several studies have focused mainly on the wood-destroying potential and lignocellulose degradation mechanisms of xylotrophic WRF belonging to the Polyporales order including such families and genera as *Polyporaceae* (*Trametes*, *Dichomitus*, *Polyporus* spp.), *Phanerochaetaceae* (*Bjerkandera*, *Phanerochaete* spp.), etc. [7,8,9,10,11]. Among them, *Trametes versicolor* (L.) Lloyd and *Phanerochaete chrysosporium* Burds. species are used as reference organisms to study complex degradation mechanisms of plant biomass [12]. The lignocellulolytic system of these WRF species has been extensively studied, and their genomes contain a large repertoire of enzymes such as cellulases, hemicellulases and ligninases that degrade cellulose, hemicellulose and lignin, respectively [7,13].

However, WRF of the order Agaricales [14], many of which are litter saprotrophs (i.e., a key ecological group involved in forest litter decomposition), are also of increasing interest actually [15,16]. Although the lignocellulose destruction strategy of the fungal group is less well understood, it is known that they act on partially decomposed plant substrates that are more resistant to biodestruction [17,18]. Nevertheless, it is known that the genomes of litter saprotrophs (especially those of the order Agaricales) contain an increased number of genes encoding oxidative enzymes such as laccases (Lacs; EC 1.10.3.2), aryl-alcohol oxidases (AAOs; EC 1.1.3.7), lytic polysaccharide monooxygenases (LPMOs; EC 1.14.99.-) and unspecific peroxygenases (UPOs, EC 1.11.2.1) [14]. At the same time, a number of class II peroxidases (PODs) genes, such as lignin peroxidases (LiPs; EC 1.11.1.14), manganese peroxidases (MnPs; EC 1.11.1.13) and versatile peroxidases (VPs; EC 1.11.1.16), decrease [19]. Laccase and ligninolytic peroxidases (PODs) are the main lignin-degrading enzymes secreted by most WRF of the order Polyporales, including well-known fungal genera such as *Polyporus*, *Lentinus*, *Pycnoporus*, *Trametes* and others. After lignin degradation, cellulose and hemicellulose fragments are released to the surface and are degraded by a wide range of cellolytic enzymes, including various hydrolases (cellulases and hemicellulases) and oxidative enzymes, including LPMOs. If the sugars formed as a result of hemicellulose degradation are a source of food (carbon) for fungi, the products of lignin degradation may be toxic to them. Both fungal ligninolytic enzymes (Lacs and PODs) and intracellular cytochrome P450 monooxygenases are involved in their detoxification. Nonspecific peroxygenases belonging to the heme protein superfamily have been discovered and characterized in members of the order Agaricales, including genera such as *Cyclocybe* (syn. *Agrocybe*), *Marasmius*, *Coprinellus*, *Coprinopsis* and *Candolleomyces* (*Psathyrella*) [20]. UPOs are oxidoreductases that act “with H_2_O_2_ as an acceptor, one oxygen atom of which is incorporated into the product” and are considered to be extracellular analogs of intracellular P450 monooxygenases, both in terms of the reactions catalyzed and their catalytic versatility. They oxidize a wide range of organic substrates, including alkanes/alkyls, cycloalkanes/cycloalkyls, aromatics/phenyls, heterocycles, halides, etc., by performing different types of oxygenation reactions—hydroxylation, epoxidation, dealkylation, deacylation and aromatization [20]. Since PODs and UPOs require the presence of hydrogen peroxide in the environment to function effectively, WRF can also secrete various enzymes such as aryl alcohol oxidases, glucose oxidase and others that produce H_2_O_2_ as a result of the oxidation of sugars and various small organic molecules released during lignocellulose degradation.

Among the WRF of the order Agaricales, the family *Nidulariaceae* is still poorly understood [21]. Within the family, most studies have traditionally focused on the genus *Cyathus* [22,23,24], while other species of the family still remain poorly understood.

The basidiomycete *Crucibulum laeve* (Hudson) Kambly is a white rot-causing gasteromycete saprotroph (order Agaricales, family *Nidulariacea*e). It is a common organism on decaying woody plant debris on the forest floor (e.g., stumps, dead fallen trunks and branches) [25]. It is known that this species can accelerate decomposition of polycyclic aromatic hydrocarbons (PAHs) [26,27], and its ability to destroy other xenobiotics (e.g., phthalic acid esters) has also been reported [28]. However, reported data on the wood-destroying capacity of *C. laeve*, as well as those on its lignocellulolytic enzyme complex, are currently limited.

To the best of our knowledge, this study represents the first attempt to characterize the saprotrophic wood-degrading abilities of a WRF *C. laeve* LE-BIN 1700 (Agaricales, Nidulariaceae) from the Komarov Botanical Institute Basidiomycetes Culture Collection (LE-BIN, St. Petersburg, Russia) isolated from twig (Western Caucasus, Adygei Republic, Russia).

We have previously shown in biochemical plate tests that this *C. laeve* isolate differs from other tested white rot fungi in its enzymatic activity spectrum [29]. In these tests, two substrates (ABTS and Azur B) were used to assess the overall oxidative capacity of the fungal mycelium, and carboxymethyl cellulose (CMC) was used as a substrate to assess the overall cellulolytic activity. While ABTS is considered to be an easily degradable substrate that can potentially be oxidized by Lacs and PODs, Azur B can only be degraded by enzymes with high redox potential, such as ligninolytic peroxidases. The *C. laeve* isolate LE-BIN 1700 demonstrated a high capacity to degrade ABTS, while the capacity to degrade Azur B and CMC was not detectable.

Here, we provide insight into the ability of the forest floor fungus *C. laeve* to degrade birch, alder and pine sawdust. Using a combination of biochemical assays, proteomics, metabolomics and genomics, we analyzed the ability of the fungus to degrade different lignocellulosic substrates.

## 2. Materials and Methods

### 2.1. Fungal Strains and Inoculum Production

In the fungal strain of *Crucibulum laeve* (Huds.) Kambly, 1936 were isolated (17 September 2003) from basidiospores collected from a fallen dead branch in the fir–beech forest (Western Caucasus, Adygei Republic, Kavkazsky nature reserve, vic. Guzeripl reserve station, Russia; N 43°59′; E 40°07′). After morphological and genetic verifications, the strain was deposited in the Komarov Botanical Institute Basidiomycetes Culture Collection (LE-BIN; St. Petersburg, Russia) as *C. laeve* LE-BIN 1700. The sequence of its ITS1-5.8S rRNA-ITS2 region is available at the NCBI GenBank accession MK795850. The fungal strain *Trametes hirsuta* LE-BIN 072 was also obtained from LE-BIN (GenBank accession number: MK795848) [29]. In the laboratory, the fungal strains were stored on slanted wort agar at 4 °C.

To prepare the inoculum, fungal strains were statically cultured in 750 mL Erlenmeyer flasks with ceramic beads and containing 200 mL of glucose–peptone (GP) medium with the following composition (g/L): 3.0 peptone; 10.0 glucose; 0.6 KH_2_PO_4_; 0.4 K_2_HPO_4_; 0.5 MgSO_4_ × 7H_2_O; 0.05 CaCl_2_; 0.05 MnSO_4_; 0.005 FeSO_4_ and 0.001 ZnSO_4_. Cultivation was carried out at 25 °C in the dark until generation of floating “mat of mycelium” covered the entire surface of the medium. To prepare the inoculum, the mycelial mat was homogenized with ceramic beads under aseptic conditions. For this purpose, the flasks containing the mycelium and beads were shaken at 250 rpm for 15 min. All the inoculations were made on 25 mL of disrupted mycelium.

### 2.2. Growth Rate Study in Petri Dishes

Growth rates of the *C. laeve* on agar media supplemented with different substrates such as sawdust of different types of wood (birch, alder and pine), and with individual lignocellulose components were studied in vitro using 50 g/L malt extract agar (MEA; Conda, Madrid, Spain) and 20 g/L agar (Difco, Kansas City, MO, USA), respectively. An agar plug (Ø = 7 mm) of an actively growing fungal mycelium was placed in the center of a Petri dish (Ø = 90 mm) containing the appropriate substrate. Cultivation was carried out at 25 °C in the dark.

Birch, alder and pine sawdust was pre-crushed in a laboratory mill IKA M20 (IKA, Werke Staufen, Germany) and then sieved (fraction size—1 mm). Then, 15 g/L of crushed sawdust was added to the MEA medium, autoclaved and dispensed sterile in Petri dishes. Control platings were carried out on a sawdust-free MEA medium.

Carboxymethyl cellulose (CMC, Sigma, St. Louis, MO, USA); birch and larch xylan (Sigma, St. Louis, MO, USA); lignosulfonate (Sigma, St. Louis, MO, USA); starch (MP Biomedicals, France) and pectin (Shanghai Acmec Biochemical, Shanghai, China) were used as separate components of lignocellulose (1 g/L).

Three replicates were made for each medium sample. Colony growth diameter measurements were taken daily for each replicate in three directions. Mean values were calculated.

### 2.3. Oxidative and Cellulolytic Plate-Test Activity by Agar Diffusion Method

Agar plugs (Ø = 7 mm) were cut from completely covered Petri dishes with different types of sawdust (see Section 2.1) and transferred to new Petri dishes (Ø = 90 mm) containing 20 mL of medium consisting of 1% (*w*/*v*) agar and 0.1% (*w*/*v*) 2,2′-azino-bis 3-ethylbenzothiazoline-6-sulfonic acid (ABTS; Sigma, St. Louis, MO, USA) or CMC to assess oxidative and cellulolytic activities, respectively. Incubation was performed in the dark at 25 °C for 48 h. Oxidative activity was assessed by the diameter of the ABTS staining zones around the fungal mycelium agar plug. Cellulolytic activity was assessed by the diameter of the enlightenment zone around the fungal mycelium agar plug. The zone of enlightenment was revealed by treatment with solution of 0.5% I in 2.0% KI.

### 2.4. Semi-Solid Cultivations, Enzyme Activity and Biochemical Assay

To characterize fungal exoproteomes and metabolomes, the semi-solid cultivations were performed in 750 mL Erlenmeyer flasks containing 200 mL of GP medium and 5 g of either alder, birch or pine sawdust; control cultivations were performed on GP medium without sawdust. Sawdust was obtained as described in Section 2.2. Each flask contained a floating nylon mesh disk to support superficial mycelial growth. The flasks were cultured statically in the dark at 25 °C for 30 days.

During cultivation, culture liquids (CLs) were sampled to measure oxidative, cellulase and hemicellulase enzymatic activities, using ABTS, Azo-CMC and Azo-xylan as substrates, respectively, as described in [30].

At the end of cultivation, the fungal biomass was separated by filtration and dried at 60 ± 5 °C to constant weight. In aliquots of the CLs, the total free phenolic content and the total amount of reducing sugars were determined using the Follin–Ciocalteu [31] and Nelson–Somogyi assays, respectively, as described in [32]. Esterase and lipase activities were determined using *p*-nitrophenyl butyrate (Sigma-Aldrich, MO, USA) and *p*-nitrophenyl palmitate (JHChem, Hangzhou, China), respectively, as described in [33], except that the *p*-nitrophenyl palmitate was dissolved in chloroform. The reaction was carried out in sodium acetate buffer (pH = 4.5) at 40 °C for 10 min. The reaction was quenched with sodium phosphate buffer (pH = 7.3) and the optical density was determined at λ = 400 nm. Esterase activity was calculated using the formula:A (U/mL) = 0.13 × ΔA400 × R E,(1)
where RE is the preliminary dilution of CL before adding the substrate to the solution.
ΔA400 = A400 − A400(S) − A400(E),(2)
where A400(S) is the control, in which water was used instead of CL; A400(E) is the control without adding a substrate to the reaction mixture.

### 2.5. Exoproteomic Study

Samples of CLs pre-filtered through a 0.45 µm membrane filter (Millipore, Burlington, MA, USA) were desalted and concentrated in a Labscale TTF system using a Biomax 5 membrane (Millipore, Burlington, MA, USA). Proteins were precipitated by adding an equal volume of a solution consisting of 13.3 mL of trichloroacetic acid (TCA) and 93 µL of β-mercaptoethanol in 100 mL of acetone. Two-dimensional (2D) gel electrophoresis of the proteins was performed using Protean II xi 2-D Cell (Bio-Rad, Hercules, CA, USA) as described in [30]. The resulting protein maps were examined on an Ultraflex II MALDI TOF/TOF mass spectrometer (Bruker, Bremen, Germany). Peptide mass and MS/MS data were analyzed using Mascot Server ver. 2.6.

### 2.6. Metabolomic Study

To perform a metabolomic study, samples of culture liquors (CLs) (15 mL) were freeze-dried, dissolved in pyridine and, then, TMS (trimethylsilyl) derivatives were obtained using *N*,*O*-Bis(trimethylsilyl)trifluoroacetamide (BSTFA, Sigma-Aldrich, St. Louis, MO, USA). Silylation was performed at 100 °C for 15 min.

Metabolites were determined in CLs via gas chromatography–mass spectrometry (GC-MS) on a GC-MS QP 2010 Ultra EI device (Shimadzu, Japan) equipped with an autoinjector and a quadrupole mass spectrometer detector. Data collection and chromatogram processing were performed using LabSolutions GCMSsolution software ver. 4 (Shimadzu, Kyoto, Japan). The relative intensities (hereafter—relative content) of individual compounds were calculated by normalizing the total intensity of the identified peaks. Compounds were identified by comparing their experimental spectra with those of the National Institute of Standards and Technology (NIST/EPA/NIH mass spectrum database, NIST 11). Fungal inoculum-free media were used as the controls.

Column—MDN-5 (30 m × 250 µm × 0.25 µm; Supelco, Bellefonte, PA, USA); mobile phase—helium, eluent flow rate—1 cm^3^/min, split ratio—1:5. Injection volume—1 µL; oven temperature—120 °C; injector temperature—200 °C. Temperature gradient: 120 °C, hold up time—1 min; from 120 to 280 °C at 10 °C/min, isotherm—3 min.

The retention index (RI) for each compound detected was calculated by normalizing its retention time (RT) by the RT and RI of the most closely eluting n-alkane, included in the standard mixture (Sigma-Aldrich, MO, USA) of even n-alkanes (C8-C32). This mixture was analyzed at the beginning of each sequencing run.

### 2.7. Wood Decay Study

The degradation capacity of the *C. laeve* and *T. hirsuta* was tested according to a modified EN 113-2 (2021) [34] test protocol using birch, alder and pine sawdust. All wood sawdust was dried to a constant weight at (103 ± 2) °C prior to testing. A total of 10 g of each sawdust type was placed in 750 mL Erlenmeyer flasks to which 10 mL of tap water was added. After double autoclaving (1 atm for 30 min), *T. hirsuta* or *C. laeve* inoculum was introduced. The fungal inoculum was obtained as described in Section 2.1. The inoculated flasks were placed in an environmental chamber (TXB-500, NPF Termokon, Moscow, Russia) and incubated in the dark at (23 ± 2) °C and 70% relative humidity [35]. At the end of the incubation period (60 and 120 days for *T. hirsuta* and *C. laeve*, respectively), sawdust was removed from the flasks and dried in a hot air oven at 105 °C to constant weight. After weighing the dry sawdust before and after fungal cultivation, the percentage of mass loss due to fungal decay (ML) was calculated using the following equation:ML (%) = (*m*_0_ − *m*_0,inc_)∕*m*_0_ × 100,(3)
where *m*_0_—the oven-dry mass before incubation [g] and *m*_0,inc_—the oven dry mass after incubation [g].

Additionally, check test flasks were implemented for every type of wood sawdust setting to measure mass loss that was not caused by fungal decay (EN 113-1 2021 [36]).

### 2.8. Statistical Data Analysis and Bioinformatics

The genome sequence of *C. laeve* was obtained from the NCBI database (accession number GCA_004379715.1). For metabolic pathways reconstruction, EC numbers were extracted from genome annotations and were automatically mapped to the Kyoto Encyclopedia of Genes and Genomes (KEGG) metabolic pathways using KEGG Mapper (https://www.genome.jp/kegg/mapper/color.html, accessed on 11 September 2024).

Proteins found in the fungal secretomes were analyzed using the NCBI conserved domain database [37], and SignalP 6.0 Server was used to predict the signal peptide [38]. Amino acid sequence alignments and visualizations were performed using CLC Main Workbench 5.5.

All the cultivations were performed in three biological replicates. For the analysis of exoproteomes and metabolomes, culture liquids from biological replicates were pooled together. Whenever appropriate, the measurement was performed in three technical replicates. All statistical comparisons were firstly performed using an ANOVA omnibus F-test. When the omnibus test demonstrated the presence of significantly different means *p* < 0.05), the ANOVAs were followed by Tukey’s honestly significant difference (HSD) post hoc tests (*p* < 0.05). Results are presented as the mean ± standard deviation (SD).

## 3. Results

### 3.1. Lineal Growth, Oxidative and Cellulolytic Activities During Cultivation on Solid Agar Media Containing Various Types of Sawdust

The WRF *C. laeve* LE-BIN 1700 showed the same linear growth rate (on average, about 9.0 mm/day) on MEA medium supplemented with birch sawdust (MEA-B) and alder sawdust (MEA-A). The fungal growth rate on MEA supplemented with pine sawdust (MEA-P) was 1.5 times lower (approximately 6.0 mm/day) (Figure 1A). Enzyme activity testing using the application method showed that both total oxidative and carboxy methyl cellulase (CMCase) activities were induced on media containing all the sawdust types (Figure 1A,B). At the same time, the activity values measured on the sawdust-containing media did not differ significantly between the different wood species (*p* > 0.05).

### 3.2. Lineal Growth and Oxidative Activity During Cultivation on Solid Agar Media Containing Different Lignocellulosic Components

*C. laeve* LE-BIN 1700 demonstrated growth efficiency on an agar medium containing model lignocellulose components (i.e., lignin, CMC, xylan, starch and pectin). In addition, lignocellulose components added to the agar significantly (*p* < 0.05) increased the fungal growth rate as compared with the control agar medium (Figure 2). Fungal mycelium showed similar growth rates (about 10.0 mm/day) on media containing lignin, xylans and CMC, but was slightly higher (about 12–13 mm/day) on media containing starch and pectin.

### 3.3. Characterization of Enzymatic Activities, Mycelial Biomass and Acidity During Semi-Solid Cultivation on Media Containing Various Types of Sawdust

During the semi-solid cultivation of *C. laeve* 1700 on the control GP medium and on the GP media containing birch, alder and pine sawdust (GP-B, GP-A and GP-P), oxidase, cellulase and hemicellulase enzymatic activities were measured in the culture liquid (Figure 3A). At the end of the 30-day cultivation, the pH value was measured and the amount of fungal biomass was determined (Figure 3B).

During the cultivation of *C. laeve* 1700, cellulase and hemicellulase activities were not observed on the substrates (Azo-CMC, Azo-Xylan, Azo-Xyloglucan and Azo-Arabinoxylan). However, oxidase activity measured on the ABTS substrate was detected on all the media (Figure 3A). *C. laeve* cultivation showed the lowest oxidative activity on the GP medium; the highest activity (28 U/mL) was reported on the 10th day of the fungal culture growth. Oxidase activity was significantly induced in the sawdust-containing media. On the 10th and 25th day of the fungal culture growth, maxima of oxidase activity (84 and 114 U/mL, respectively) were observed on GP-A and GP-B media (Figure 3A). On GP-P medium with pine sawdust, the activity gradually increased and reached a value of about 170 U/mL on the 30th day of cultivation.

Compared to the GP medium, the growth of the fungal biomass increased significantly (*p* < 0.05) on the sawdust-containing media; the maximum value was recorded on the GP-B medium with birch sawdust (Figure 3B). During its growth on the all media, *C. laeve* slightly acidified the culture liquids. In particular, the pH values were on average 4.8 ± 0.2 units compared to 5.4 ± 0.3 units measured at the beginning of the cultivation.

### 3.4. Exoproteomes Analysis During Semi-Solid Cultivation on Media Containing Various Types of Sawdust

In total, 61 proteins (Figure 4, Appendix A) were found on 2DE maps of *C. laeve* exoproteomes; 17–37% of them were proteins with unknown function, and the largest number was detected on GP-P (Figure 5). Of the total secretome protein pool, 35–49% were carbohydrate-active enzymes (CAZymes) [39]. At the same time, 50% of the total CAZyme pool on the GP medium were glycoside hydrolases (GHs); auxiliary activities (AAs) and carbohydrate esterases (CEs) classes were 37% and 13%, respectively. The ratio changed on the sawdust-containing media: GHs content decreased; AAs content, on the contrary, increased; CEs: 6–8% (GP-A and GP-P) and 22% (GP-B) (Figure 5).

The *C. laeve* GP exoproteome is represented by the largest repertoire of proteins (*n* = 53). Approximately one-third (*n* = 12) of these proteins were secreted on all the media, another third (*n* = 12) were unique to medium, and the remaining proteins were secreted in different ways on the sawdust-containing media. The major proteins found on all the media were laccases (AA1_1), LPMOs X325 family (Appendix A), GMC oxidoreductases (AA3) and FAD-binding domain-containing protein that was assigned to the AA7 CAZyme family (EC 1.1.3.-; oligosaccharide oxidase), as well as expansin-like proteins like fungal loosenins (Appendix A) and lectins (Appendix A, Figure 4). Their production was significantly increased on the sawdust-containing media. At the same time, the sawdust-containing media showed a decrease in the total number of proteins as compared to the GP medium (total 32, 28 and 22 proteins on GP-B, GP-A and GP-P media, respectively). A similar trend was also observed for other WRF (*Trametes hirsuta*; *Steccherinum ochraceum*; *Peniophora lycii*) when cultivated under similar conditions [29,30].

The GHs repertoire in *C. laeve* exoproteomes presented GH16, GH17, GH18, GH31, GH37, GH47, GH51, GH72 CAZy families, among which only CAZy GH families 31 and 51 are involved in the biodegradation of plant cell walls. Among the CEs in the *C. laeve* exoproteome, representatives of only two families (CE4 and CE16 CAZy) were found.

TFK32449.1 and TFK42464.1 (proteins of the carboxylesterase/lipase family) were detected among the “Others” proteins in the *C. laeve* secretomes. Esterase and lipase activities measured in culture liquids showed that esterase activity was higher than lipase activity on all the media, while both activities increased on the sawdust-containing media in contrast to those on the GP medium (Table 1). The highest esterase and lipase activities were found on the GP-A and the GP–B media, respectively.

Proteases of different families as well as a large number of proteins with unknown functions were also found in the secretomes of *C. laeve* (Figure 4). In the case of the latter, these are mainly low molecular proteins (10–25 kDa) with a large number of serine/glycine/cysteine repeats that is typical for various adhesins and hydrophobins.

### 3.5. Preliminary Metabolome Analysis During Semi-Solid Cultivation on Media Containing Various Types of Sawdust

At the end of *C. laeve* cultivation, the content of reducing sugars on the GP and the GP-P media was lower than initial sugar concentration (Table 2). On the contrary, after the fungal cultivation, the reducing sugars content on the GP-B and the GP-A media was higher than at the beginning of cultivation.

Preliminary GC–MS metabolic analysis was carried out using the standard derivatization scheme which showed that the composition of *C. laeve* metabolomic profile was dependent on a culture medium (Appendix A). It was found that the major low molecular weight metabolites in the fungal CLs samples were, predominantly, monosaccharides and carboxylic acids. Notably, the levels of monosaccharides and polyols in a total pool of identified metabolites increased on the sawdust-containing media (Figure 6, Appendix A). Glucose, galactose, mannose, xylose, arabinose, rhamnose and ribose were detected on the sawdust-containing media. The GP-B medium also contained the disaccharides α-(1→2)– and α-(1→3) mannobiose, which are part of the fungal cell wall. Among the polyols identified, glycerol and erythritol were the predominant compounds on the sawdust-containing media; 1,5-anhydro-D-sorbitol was additionally detected on the GP-B medium. Lactic acid was also detected on all the media and glycolic (2-hydroxy acetic) acid was identified on the media containing alder and pine sawdust. Although no amino acids were found among the metabolites, hydroxy fatty acids, i.e., intermediates in the metabolism of branched-chain amino acids—alpha-hydroxyisovaleric acid on the GP medium and 3-hydroxyisobutyric acid or 3-hydroxy-2-methylpropanoic acid—were detected on all the sawdust-containing media.

Among the phenolic compounds, benzeneacetaldehyde was found on the GP medium only, while at the end of the fungal cultivation, the total phenol content on all the media was significantly lower than that on the initial ones (Table 2).

### 3.6. Wood Degradation Analysis During WRF Crucibulum laeve LE-BIN 1700 and Trametes Hirsuta LE-BIN 072 Solid-State Cultivation on Various Types of Sawdust

Among WRF wood–destroying Basidiomycetes, *Trametes* spp. (*Polyporaceae* family) are reference organisms able to efficiently biodegrade all wood components (lignin, cellulose and hemicelluloses) [40]. On the contrary, the ability of white rot fungi (*Nidurulaceae* family) to destroy the complex structure of plant biopolymers is still under investigation. In order to investigate the saprotrophic wood decomposition ability of *C. laeve* (*Nidurulaceae* family) and to compare it with that of the primary wood xylotroph *T. hirsuta*, a potent lignocellulose destructor [41], solid-phase cultivation of both fungal cultures was carried out on different wood sawdust (birch, alder and pine). The cultivation time of *T. hirsuta* on all the types of sawdust was 60 days; as for *C. laeve*, the cultivation time was extended to 120 days in proportion to a growth rate difference between these two fungi, since we had previously shown that the growth rate of *T. hirsuta* mycelium on agar media containing birch, alder and pine sawdust is about 16–18 mm/day [30], whereas that of *C. laeve* is about two times lower (6–9 mm/day) under the same experimental conditions (Figure 1A).

Analysis of the mass loss (Figure 7A) and specific gravity variation (Figure 7B) in sawdust after cultivation of the WRF *T. hirsuta* and *C. laeve* (Figure 7 top and bottom, respectively) showed that the primary wood xylotroph (*T. hirsuta*) degrades hardwood sawdust more strongly than coniferous sawdust. For example, mass loss of birch and alder sawdust was approximately 25%, and their specific gravity decreased by 32 and 43%, respectively (Figure 7 top). At the same time, the loss of mass and specific gravity of pine sawdust was 10–13%. As for the *C. laeve* saprotroph, the highest mass loss and specific gravity loss (about 23%) were detected in birch sawdust. The fungus decomposed alder and pine sawdust at the same rate (the loss of mass and specific gravity of alder and pine sawdust were 10–15%), but more slowly than birch sawdust (Figure 7 bottom).

## 4. Discussion

Naturally, wood-destroying and litter saprotrophic fungi are primarily involved in the degradation of lignocellulose [3]. Indeed, WRF possess a powerful enzymatic apparatus necessary for the complete degradation of all the plant cell wall components (i.e., cellulose, hemicellulose and lignin). Currently, the enzymatic system that destroys the cell wall of plants has been best studied in wood-destroying fungi WRF (Polyporales, *Trametaceae*) [7,8,9,10,11]. In contrast, the molecular mechanism of lignocellulose destruction in Basidiomycetes living on forest litter (litter rot) is less understood, although they are known to act on partially destroyed (decomposed) plant biomass, which is extremely resistant to enzymatic destruction [17,18]. In this paper, we evaluated the efficiency of the WRF *C. laeve* (Agaricales, *Nidulariaceae*) in destroying birch, alder and pine sawdust, and also analyzed the secretomic and metabolic profiles resulting from the fungal cultivation on the sawdust-containing media.

### 4.1. Growth and Wood Degradation Ability

*C. laeve* LE-BIN 1700 demonstrated the ability to grow and to destroy birch, alder and pine sawdust (Figure 1A and Figure 7). The fungal mycelium completely colonized the Petri dishes in 10 days of growth on the birch and alder sawdust agar media, while the growth rate was about two times lower on the pine sawdust media (Figure 1A). *C. laeve* demonstrated the ability to grow on agar media with different lignocellulosic components (i.e., lignin, CM cellulose, xylane, starch and pectin), and the highest growth rates were reported on media with starch and pectin (Figure 2). Under the same conditions, *T. hirsuta* LE-BIN 072, a wood-destroying white rot fungus, showed approximately 2.5–3 times higher growth rates [30] than those of *C. laeve*. Fast-growing white rot fungi are known to cause more rapid wood decay than slow-growing fungi [42]. Therefore, the decrease in weight and the specific gravity of sawdust during the cultivation of the fast-growing *T. hirsuta* and the slower-growing *C. laeve* was evaluated in 60 and 120 days, respectively. At the same time, both fungi showed comparable weight loss and reduced specific gravity of sawdust at the end of cultivation (Figure 7). A distinctive feature of the *C. laeve* was a significantly higher degradation and saccharification capacity on birch sawdust compared to alder and pine, which in turn correlated with a large increase in biomass during semi-solid cultivation of fungus on birch sawdust (Figure 3B). The *T. hirsuta* was characterized by a more pronounced destruction of deciduous sawdust (i.e., birch and alder) as compared to coniferous one (pine sawdust) (Figure 7). Coniferous wood has, as usual, lower decay rates than deciduous wood [43]. Wood decay rates are significantly influenced by the chemical composition (content and composition of extractive compounds) and/or structural characteristics of the wood (composition and cellulose/hemicellulose/lignin ratio). A strong negative correlation between wood decay rate and extractive compound content was reported in [43]. Analysis of wood chemical composition showed that alder wood has the highest extractive compound content (6.6%) compared to birch and pine wood (2.9% and 5.2%, respectively) [44]. Each tree species is characterized by a unique class of extractive compounds. For example, alder extracts contain various diarylheptanoids and tannins, whereas pine extracts contain stilbenes, terpenoids and other compounds [45,46]. According to [44], alder wood has a lower cellulose and hemicellulose content (44.1% and 33.1%, respectively) and a higher lignin content (22.0%) than birch wood (45.4%, 38.8% and 17.7%, respectively). It is noteworthy that it was the secretome of the alder sawdust medium (GP-B) that contained a significant amount of aryl alcohol oxidases (AA3_2 CAZy), which were absent in the other media (Figure 4, Appendix A).

### 4.2. Secretion of Enzymes Related to Degradation of Lignocellulose

The degradation efficiency of the wood substrates is determined by the action of fungal enzyme systems and depends on their qualitative and quantitative composition. The different types of lignocellulose destruction patterns observed in fungi are mediated by the presence in their genomes of gene families encoding carbohydrate-active enzymes (CAZymes), mainly glycosyl hydrolases (GH) and the auxiliary activity (AA) family [12]. In this regard, the *C. laeve* genome is similar to that of the WRF *Trametaceae* family, with a complete set of CAZyme families associated with the degradation of all plant cell wall components [29]. A unique difference is the presence of a large number of gene copies—52, 34, 15 and 10 versus (15–21), (15–18), (6–7) and (2–3) belonging under AA1_1, AA3_2, AA7 and AA9 CAZyme families in the *Trametes* sp. Genomes, respectively [29].

It is known that the production of WRF enzymes is determined both by their taxonomic position and ecological niche [47]. Xylotrophic WRF of the order Polyporales, in particular representatives of the genera *Polyporus* [11], *Pycnoporu*s [10,48,49], *Dichomitu*s [50] and *Trametes* [51], when cultivated on media with sawdust, secrete a significant amount of typical cellulolases of the GH5, GH6 and GH7 CAZy families and hemicellulases of the GH2, GH3, GH5, GH10, GH11 and GH28 CAZy families, as well as cellulose degrading enzymes such as lytic polysaccharide monooxygenases (LPMOs) and cellobiose dehydrogenase (AA3). For lignin degradation, WRF secrete various ligninolytic enzymes (LDEs), including laccases (AA1) and class II peroxidases (PODs, AA2) [52,53]. However, different Polyporales species exhibit different patterns of LDE secretion: (i) secretion of Lac and PODs (MnP, LiP and VP)—*Trametes versicolor*; (ii) secretion of Lac and one of the PODs (MnP, or LiP, or VP)—*Ceriporiopsis subvermispora*; and (iii) secreting only PODs—*Phanerochaete chrysosporium* [54]. WRF of the order Agaricales, in particular the genera *Schizophyllum*, *Agaricus* and *Cyclocybe* (=*Agrocybe*), whose members are litter-degrading or humus (soil) saprotrophs when grown on media with lignocellulosic substrates, also secrete cellulases and hemicellulases, but a wider range of hemicellulases hydrolyze the side chains of polysaccharides (family GH27, GH31, GH35, GH43, GH51, GH115, CE1, CE15 CAZy). However, unlike Polyporales, the secretomes of WRF Agaricales lack PODs, while some species secrete laccases (*Agaricus bisporus*), and others do not (*Schizophyllum commune*). Another distinguishing feature of the secretomes of WRF Agaricales from Polyporales is the high diversity of LPMOs, glucooligosaccharide oxidases (AA7 CAZy) and expansin-like proteins [55,56,57]. Some WRF Agaricales species show less cellulose degradation and more lignin degradation, while others in contrast show more intense cellulose degradation. Therefore, some species of this taxonomic group were not susceptible to the WR/BR classification and occupy an intermediate position between white and brown rot fungi—*S. commune* and *Armillaria* spp. [58,59]—or between white and soft rot fungi, as in the case of *Armillaria ostoyae* and *Armillaria cepistipes* [60], and the species *Mucidula mucida* (= *Oudemansiella*) is capable of causing both soft and white rot depending on the stage of decomposition and the type of wood substrate [59,61].

Similarly to other WRF of the order Agaricales, oxidative enzymes such as lytic polysaccharide monooxygenases (LPMOs), laccases and oxidoreductases, families AA3_2 and AA7 CAZymes and also expansin-like proteins were identified in the *C. laeve* secretomes. Moreover, their secretion was increased manifold on the sawdust-containing media and additional isoenzymes appeared (Figure 4, Appendix A). At the same time, *C. laeve* secretomes lacked the major cellulose- and hemicellulose-degrading enzymes (GH1, GH2, GH3, GH5, GH10 and GH11 CAZy). It was previously thought that oxidative enzymes were only involved in lignin degradation. Recently, however, the degradation of cellulose and hemicellulose, in which LPMOs play an essential role, has been actively studied [62]. Today, LPMOs are grouped into eight families of CAZymes (AA9–AA11 and AA13–AA17) and have been shown to oxidatively cleave various oligosaccharides (such as cellulose, hemicellulose, pectin, starch and chitin) [63]. A more in-depth study of LPMO sequences found in the *C. laeve* secretomes revealed that they contain the DUF6595 domain (pfam20238), similarly to recently characterized members of a new LPMO family from *Laetisaria arvalis* (defined as X325), which represents a widespread family of copper-containing proteins in various fungal genomes [64]. Although the overall structural fold of the X325 proteins was similar to that of the LPMOs, they did not possess activity against the polysaccharides tested and their functions may vary in different fungi. Despite the typical conservative pattern of X325 proteins, the identity between LPMOs from *C. laeve* secretomes and X325 LPMO from *L. arvalis* (MK088083) is only about 30% (Appendix A). It is worth noting that the absence of enzymatic activity of various LPMOs is quite common under model experimental conditions. In work [65], two AA14 LPMOs (*Pc*AA14A and *Pc*AA14B) from the WRF *Pycnoporus coccineus* were characterized to be inactive against model substrates such as wheat arabinoxylan, arabinan, barley β-glucan, curdlan, konjac glucomannan, tamarind xyloglucan, starch, Avicel, chitin, laminarin, PASC and birch xylan. However, pine and poplar substrates pretreated with the *Trichoderma reesei* cellulase–xylanase cocktail were associated with an increase in saccharification ability with the addition of *Pc*AA14A and *Pc*AA14B LPMOs. It has also been shown that different LPMO isoenzymes can have different preferences for different types of oligosaccharides [66]. These studies, which indicate a close relationship between the chemical structure of plant polysaccharides and their interaction with the substrate-binding surface of LPMOs, shed light on a wide variety of LPMO genes in fungal genomes.

In addition to LPMOs, loosenin-like expansins were found in the secretomes of *C. laeve*. Moreover, increased secretion was showed for some of them on the sawdust-containing media, and also new proteins were detected. Expansin-like proteins can modify the surface of various lignocellulosic substrates in a non-enzymatic way, thus enhancing the enzymatic degradation of lignocellulose [67,68]. The synergistic effect of expansins and LPMOs has also been reported [69]. In addition, the combination of xylanase with *Cmi*EXLX2 and *Daq*EXLX1 expansins increased product yield from hardwood pulp by approximately 25%, while *Tr*AA9A LPMO from *Trichoderma reesei* supplemented with the same expansins increased total product yield by over 35%. A recent study demonstrated the differences in the specificity of the interaction of different expansins with various polysaccharides. It was suggested that the multiplicity of expansins in an organism and the fine regulation of their expression may in part be caused by the specific binding of individual expansins to particular cell wall polysaccharides [70].. This is in agreement with the differential secretion of the *C. laeve* loosenin-like expansins on media containing different types of sawdust detected in our study (Figure 4).

In addition to the specific molecular structure of the polysaccharides, LPMO activity also depends on the presence of an electron donor (i.e., the so-called reducing agent) in a reaction medium. Small organic molecules (such as ascorbic acid, cysteine, glutathione, gallic acid, etc.) synthesized by the fungus or lignin-derived during lignocellulose destruction, as well as enzymatic redox partners (flavo-enzymes AA3 and AA7 family CAZymes) can act as reducing agents for different LPMOs [71,72]. In our study, oligosaccharide oxidase (AA7 family CAZymes, TFK39177.1), whose increased secretion is reported on all the sawdust-containing media, and the GMC oxidoreductase (AA3_2 CAZy family, TFK37403.1), detected on the alder sawdust-containing medium (GP-A), can act as redox partners for LPMOs. In addition, according to another established molecular mechanism, laccase–mediator systems (LMS) release low-molecular weight phenolic compounds from lignocellulose, which are also electron donors for LPMOs [73]. Laccases were identified in the *C. laeve* secretomes and their secretion increased on the sawdust-containing media. Interestingly, the initial media with sawdust contained free phenols, whose concentration in CLs decreased significantly at the end of the cultivation (Table 2). On the one hand, phenoxy radicals resulting from laccase-induced phenol oxidation can then polymerize; on the other hand, phenols can be metabolized by the fungus.

In *C. laeve* secretomes, the lytic enzyme repertoire includes glycoside hydrolases (GH16, GH17, GH18, GH31, GH37, GH47, GH51, GH72 CAZy families) and glycoside esterases (CE4 and CE16 CAZy families), most of which are involved in the fungal cell wall remodeling. Interestingly, the CE16 enzyme (universal acetylesterases involved in hemicellulose degradation) was detected on the GP-B medium.

A wide range of differentially secreted lectins was found in the *C. laeve* secretomes (Figure 4). A recent study reported that fungi from different ecological groups have different lectomes, while the greatest diversity of lectin classes is observed in litter decomposers [74]. This suggests an important biological role for these proteins.

A protein profile similar to that of *C. laeve* secretomes was shown for the WRF *Peniophora lycii* of the order Rusullales when cultured on a medium containing wood sawdust, where hydrolytic enzymes such as cellulases and hemicellulases of the families GH2, GH3, GH5, GH6, GH7, GH10, GH11 and GH28 CAZy typical for WRF of the Polyporales order were absent; and predominantly oxidative enzymes were detected [30]. The spectrum of oxidative enzymes was represented by laccases (AA1), LPMOs (AA9), GMC oxidoreductases (AA3) and glucooligosaccharide oxidases (AA7). Currently, no information is available on the proteins of the exoproteomes of fungi of the family *Nidulariaceae* (Agaricales). However, transcriptional analysis of *Cyathus bulleri* (*Nidulariaceae*, Agaricales) showed that on wheat bran medium the highly transcribed CAZymes were the GH13 family encoding α-amylase, GT32 family encoding α-galactosyltransferase, CE4 encoding acetyl xylan esterase, AA7 family encoding glucooligosaccharide oxidase, GMC oxidoreductases (AA3) and the laccase and LPMO isoforms were also highly expressed [22]. Moderate transcription of cellulases (GH5) and hemi-cellulases (GH3, GH10, GH30 and GH43) and low transcription of PODs genes were also observed.

In summary, the proteome analysis of *C. leave* indicated that the members of the family *Nidulariaceae* may use a predominantly oxidative mechanism for lignocellulose degradation, at least at certain growth stages (Figure 8A). According to the exoproteome profile, *C. leave*, like some other saprotrophs from Agaricales, cannot be classified as a typical member of the white rot fungi. Proteins found in the *C. laeve* secretomes can be further exploited to develop new enzymatic cocktails for lignocellulose processing and possibly extracting biologically active compounds.

### 4.3. Metabolome Analysis of C. laeve Culture Liquids

The difference in the profile and ratios of the metabolites between the media was demonstrated by GC-MS analysis of silylated fungal culture liquids. Lactic acid, present in all the CLs, is a potential fermentation product because static cultivation (no agitation, 30 days) creates hypoxic conditions with low oxygen content. Under these conditions, free sugars both initially available in media and released during the sawdust saccharification can be fermentation substrates (Table 2). Finally, increased levels of reducing sugars were reported during cultivation on the GP-B and the GP-A media. In contrast, their content decreased on the GP-P. This may indicate a more intensive saccharification of birch and alder sawdust, as compared with pine sawdust, and the trend correlates with a large amount of biomass on the GP-B and the GP-A media (Figure 3B). Glycerol, erythritol and other polyols found on the sawdust-containing media (Appendix A) may be synthesized by the fungus not only as storage substances, but also in response to potential stress conditions at the end of the 4th week of cultivation. Glycerol and erythritol are known to be metabolically linked to intracellular storage carbohydrates such as trehalose or glycogen. Notably, osmotic stress caused by sugar accumulation in the culture medium leads to rapid glycerol formation, and oxidative stress (i.e., formation of hydrogen peroxide and ROS resulting from the action of oxidative enzymes) leads to erythritol formation [75].

Sugars (such as D-galactose and D-ribose) were found on all the sawdust-containing media. At the same time, the D-galactose content was significantly higher on the GP-B and the GP-A media than on the pine sawdust medium (GP-P) (17.48%, 13.03% and 7.93%, respectively) (Appendix A). L-rhamnose and arabinose were detected only on the GP-B medium; beta-D-glucose was found on the GP-B and the GP-A media; and D-xylose was reported on the GP-B and the GP-P media. However, no aldonic acids and/or their corresponding lactones were found in any of the media.

### 4.4. In Silico Analysis of Carbohydrate Metabolism of C. laeve

As a result of lignocellulose destruction, various oligosaccharides and monosaccharides are released. These are then metabolized by fungi as a carbon source. As for the reactions catalyzed by LPMOs, all these enzymes cleave a polymer chain of cellulose or hemicellulose with oxidation of either the C1 atom (EC 1.14.99.54) or the C4 atom (EC 1.14.99.56) of the glycoside ring to form aldonic acid (or its lactone) or 4-ketoaldose, respectively [76]. A fairly large group of LPMOs generate products oxidized both at the C1 and C4 positions in varying proportions. Aldonic acids (or their lactones) are also products of reactions involving AA3_2 and AA7 CAZymes. Thus, the oxidative degradation of lignocellulose involving enzymes detected in the *C. laeve* secretomes should produce predominantly oxidized and native oligo- and monosaccharides.

The *C. laeve* genome contains almost all the genes encoding enzymes involved in carbohydrate metabolism through glycolysis, the tricarboxylic acid (TCA) cycle, the glyoxylate cycle, the pentose phosphate pathway (PPP) and the D-galacturonic acid pathway (Figure 8B, Appendix A). These energy metabolism pathways are highly conserved among different fungi [77]. However, no genes encoding oxaloacetase (OXA, EC 3.7.1.1) were found in the *C. laeve* genome (Appendix A). The enzyme hydrolyses oxalacetate to oxalic acid and acetate in the TCA cycle and the glyoxylate shunt, which uses cytosolic or peroxisomal acetyl-CoA to produce oxaloacetate. Indeed, oxalic acid (oxalate) and acetate were also not detected in CLs of *C. laeve* (Appendix A). Oxalate secretion is thought to be a part of the lignocellulolytic machinery of both white and brown rots, enhancing the action of lignolytic enzymes such as MnP and LiP in white rots and intensifying the Fenton reaction in both rots [78]. Nevertheless, it has been shown that some fungi (for example, *Trichoderma reesei* and *Phanerochaete chrysosporium*), such as *C. laeve*, do not contain OXA genes in their genomes [77]. However, the *C. laeve* genome contains the protein Bicupin (oxalate decarboxylase/oxidase). Two different activities are known for members of this family: oxalate decarboxylase (EC 4.1.1.2) and oxalate oxidase (EC 1.2.3.4). Typically for fungi, oxalate decarboxylase leads to the formation of formic acid and carbon dioxide, and the oxidation of oxalic acid catalyzed by oxalate oxidase leads to the formation of carbon dioxide and hydrogen peroxide [79]. Interestingly, carbonic acid was found among the metabolites, and the peak intensity of this compound increased significantly on the sawdust medium compared to the GP medium (Appendix A).

D-galacturonic acid is the main component of pectin and an important carbon source for various microorganisms. Similarly to other fungi, D-galacturonic acid in *C. laeve* is catabolized via the non-phosphorylating pathway (M00630) to glycerol (Figure 8B). Probably, due to the rapid metabolism, galacturonic acid was not detected among the metabolites. In fact, the growth rate of *C. laeve* on the agar media with pectin was higher than on the media with CM cellulose and xylans (Figure 2).

According to the metabolic pathway analysis, *C. laeve* can catabolize both D-glucose via the glycolytic pathway (glycolysis or Entner–Doudoroff pathway) as well as D-gluconic acid resulting from D-glucono-1,5-lactone hydrolysis (Figure 8B, Appendix A). The latter, in turn, is a product of glucose oxidation by glucose oxidase (GOX, EC 1.1.3.4) or oxidative degradation of glucose-containing polysaccharides (e.g., cellulose, xyloglucan, etc.) with LPMOs and/or FAD-oligosaccharide oxidase (AA7 CAZymes) identified in the fungal secretomes. Under neutral or alkaline conditions, D-glucono-1,5-lactone undergoes spontaneous hydrolysis. As the reaction proceeds and D-gluconic acid (D-gluconate) accumulates, the medium becomes acidic. The lactones are then enzymatically hydrolyzed in the presence of lactonase (EC 3.1.1.17). After intracellular penetration, D-gluconate is metabolized via the PPP (KEEG pathway: map00030). It is noteworthy that the TFK32984.1 protein representing gluconolactonase (EC 3.1.1.17), which hydrolyzes D-glucono-1,5-lactone to gluconic acid, was detected in the GP medium where D-glucose was the only carbohydrate. In addition to gluconolactonase, seven other genes encoding the proteins TFK32977.1, TFK33000.1, TFK35654.1, TFK32978.1, TFK31125.1, TFK37132.1 and TFK32978.1 were predicted in the *C. laeve* genome. According to the COG database, this protein belongs to the YvrE superfamily (COG3386) which includes sugar lactone lactonases. In contrast, no lactonase genes were found in the WRF *P. chrysosporium* genome, and two copies were detected in the genome of *Dichomitus squalens* [77]. The presence of eight sugar lactone lactonases genes in the *C. laeve* genome, with quite low levels of amino acid sequence identity (from less than 50 to 86% identity), may indicate the preferential catabolism of various oxidized sugars resulting from the enzymatic oxidation of lignocellulose. This correlated with the detection of several genes of LPMOs, AA3_2 and AA7 CAZymes. Such special catabolic behavior, the ability to metabolize aldonic acids (such as D–gluconate), appears to confer certain environmental advantages on forest floor fungi, since the rapid oxidation of glucose and other monosaccharides released during the destruction of plant cell wall polysaccharides prevents carbohydrate use by competitive microorganisms, which are abundant in the econiche of the forest floor. Perhaps, such a longer glycolysis metabolic pathway, compared to D-glucose catabolism, results in a rather low growth rate of *C. laeve*.

The *C. laeve* genome contains all enzymes of the Leloir pathway (KEGG pathway: map00052) and some enzymes of the unphosphorylated De Lay–Doudoroff pathway of D-galactose metabolism (Figure 8B, Appendix A). Overall, at least one gene for mannose and L-rhamnose metabolism was identified in the *C. laeve* genome (Appendix A). Key genes for L-arabinose metabolism are absent from the *C. laeve* genome, in contrast to D-xylose metabolism, where some of them are present in multiple copies, in particular the genes for xylitol dehydrogenase (EC 1.1.1.9) and D-xylulose kinase (EC 2.7.1.17) (Figure 8B, Appendix A). It is possible that *C. laeve* is unable to metabolize L-arabinose and this may explain the lower growth rate on arabinoxylan agar medium than on other hemicellulose polysaccharides (Figure 2). All genes involved in D-ribose catabolism are also present in the *C. laeve* genome (Figure 8B, Appendix A). Further detailed studies of carbohydrate metabolism in *C. laeve* are required.

## 5. Conclusions

In this study, the degradation of birch, alder and pine sawdust by the WRF *Crucibulum laeve* (Agaricales, *Nidulariaceae*), a litter decomposer, was investigated. The fungal molecular mechanism used to degrade different lignocellulosic substrates was investigated using biochemical analyses as well as proteomics, metabolomics and genomics. The fungus showed the highest radial growth rate on media containing starch and pectin. Fungal exoproteome analysis showed that laccases, lytic polysaccharide monooxygenases (LPMOs), GMC oxidoreductases (AA3_2 and AA7 CAZyme families), as well as expansin-like proteins and lectins were the major secreted proteins in all the media.

Metabolomic analysis revealed that, in particular, monosaccharides and carboxylic acids were the major low molecular weight metabolites in the *C. laeve* culture liquids (CLs) under the experimental conditions. Moreover, the proportion of monosaccharides and polyols in a total pool of identified compounds increased on the sawdust-containing media. Among the polyols, glycerol and erythritol were the predominant compounds on the sawdust-containing media. Metabolic pathway analysis indicates that the fungus can catabolize D-gluconic and D-galacturonic acids and possibly other aldonic acids. This appears to provide *C. laeve* with certain environmental advantages. For example, rapid oxidation of glucose and other monosaccharides released during the destruction of plant cell wall polysaccharides prevents the use of carbohydrates by other competing microorganisms, which are abundant in forest litter.

The results of the study demonstrate the potential of *C. laeve* to degrade various lignocellulosic substrates and provide a fundamental understanding of the enzymes responsible for the degradation. As this is the first attempt to analyze a profile of secreted extracellular proteins for lignocellulose biomass degradation in a member of the *Nidulariaceae* family, it is likely to contribute to better understanding of lignocellulase biodegradation mechanisms by fungi of this family. A better understanding of the mechanisms of wood biopolymer decay by different types of wood-destroying fungi will allow the identification of new enzymes for the development of more effective preparations and/or technologies for the processing of lignocellulosic waste in the future.

## Figures and Tables

**Figure 1 jof-11-00021-f001:**
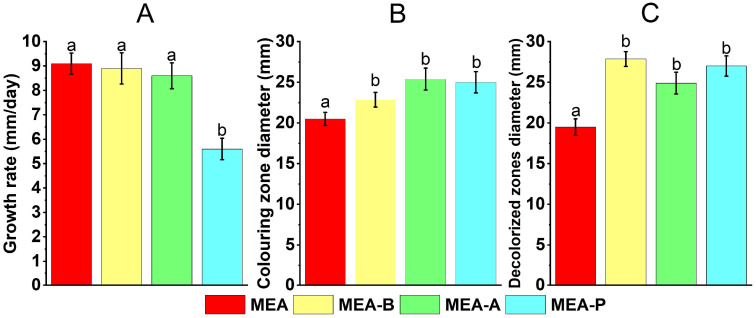
Growth rate (**A**), oxidative (**B**) and cellulolytic (**C**) activity during cultivation of *Crucibulum laeve* LE-BIN 1700 on MEA media containing various types of wood sawdust—birch (MEA-B), alder (MEA-A) and pine (MEA-P). The MEA medium without a sawdust addition was used as a control. Different letters indicate statistical differences between culture media at the same time point (*p* < 0.05).

**Figure 2 jof-11-00021-f002:**
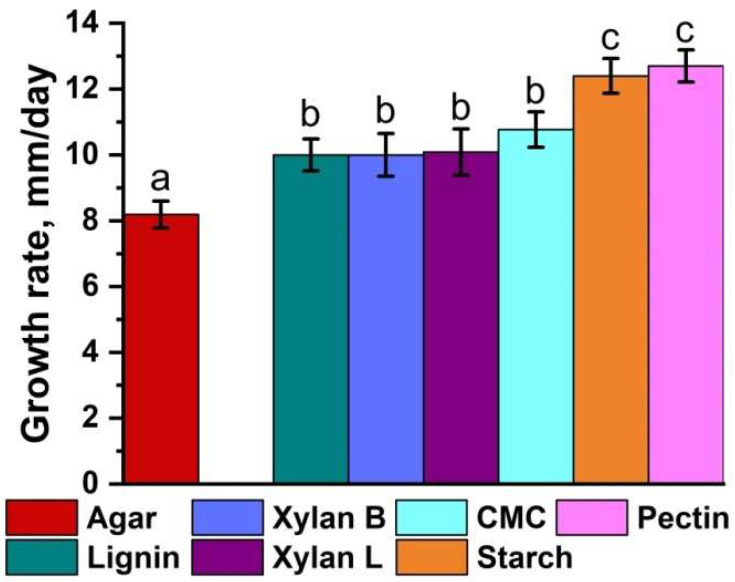
*Crucibulum laeve* LE-BIN 1700 growth rate on control medium (only bacteriological Agar) and medium supplemented with different lignocellulose compounds—lignin; birch xylan (Xylan B); larch xylan (Xylan L); carboxymetyl cellulose (CMC); starch and pectin. The same letters indicate that the values are not significantly different (*p* > 0.05).

**Figure 3 jof-11-00021-f003:**
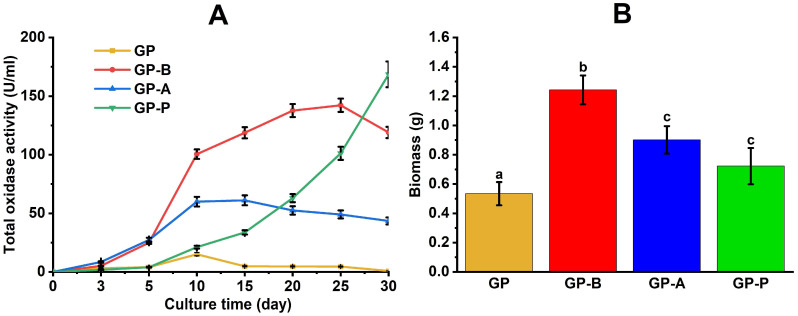
(**A**) White rot fungus (*Crucibulum laeve* LE-BIN 1700) oxidase activity variation during stationary semi-solid cultivation on the control glucose–peptone (GP) medium and on the GP supplemented with birch (GP-B), alder (GP-A) and pine (GP-P) sawdust; (**B**) fungal biomass amount at the end of 30-day cultivation. Different letters indicate statistical differences between culture media at the same time point (*p* < 0.05).

**Figure 4 jof-11-00021-f004:**
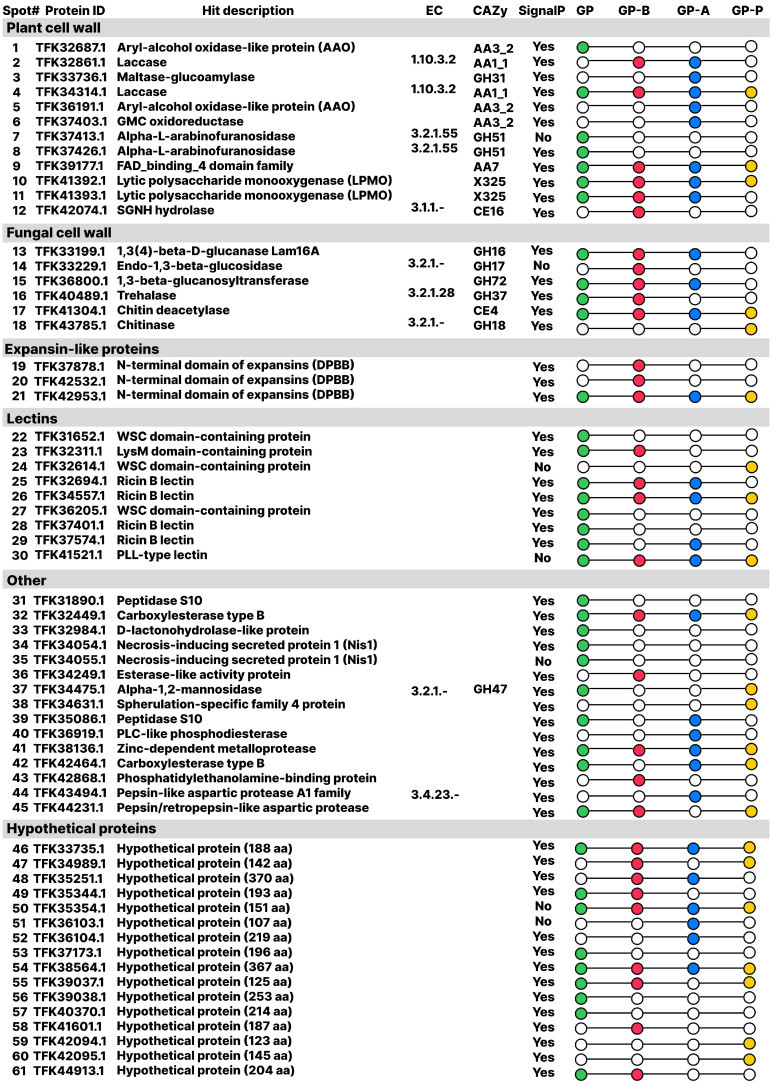
Functional description and secretion conditions of proteins found in *C. laeve* LE-BIN 1700 exoproteomes during semi-solid cultivation on the control glucose–peptone (GP) medium and on the GP supplemented with birch (GP-B), alder (GP-A) and pine (GP-P) sawdust; aa—amino acid. The presence of proteins in the corresponding medium is indicated by a colored circle.

**Figure 5 jof-11-00021-f005:**
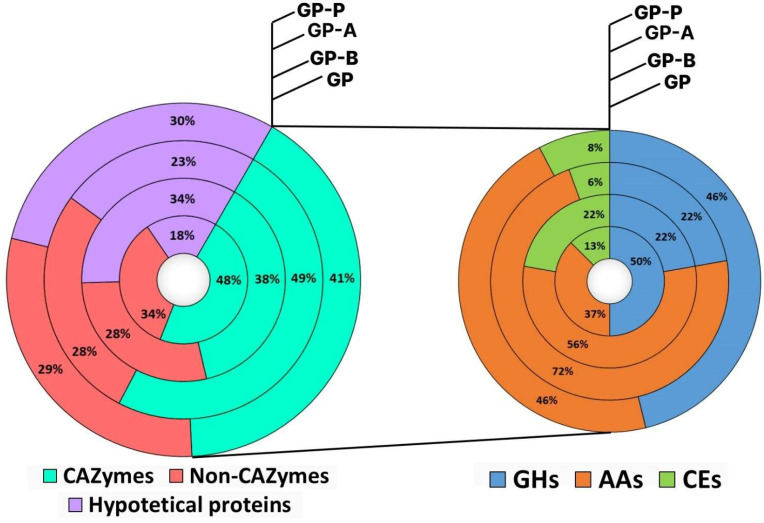
Distribution profile of identified secreted proteins and the distribution of CAZymes (GHs, AAs and CEs classes) in the GP (glucose–peptone medium), GP-B (glucose–peptone with birch sawdust), GP-A (glucose–peptone with alder sawdust) and GP-P (glucose–peptone with pine sawdust) of the *Crucibulum laeve* LE-BIN 1700 secretomes. AA—auxiliary activity enzyme; CE—carbohydrate esterase; GH—glycoside hydrolase.

**Figure 6 jof-11-00021-f006:**
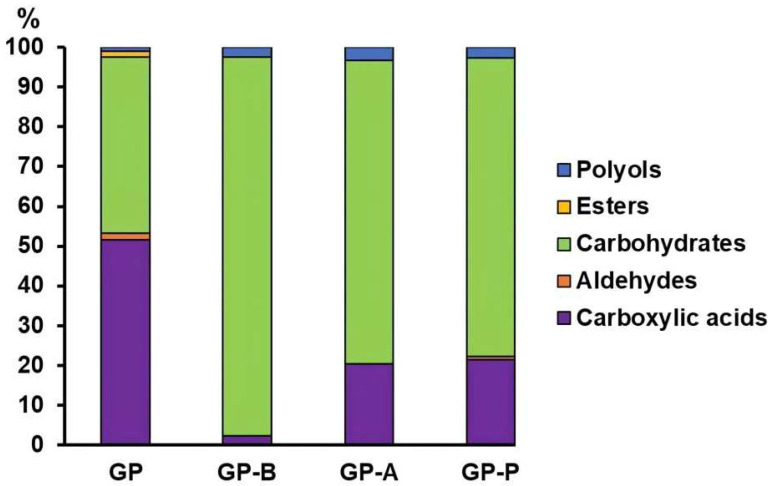
Total pool of identified metabolites in the GP (glucose–peptone medium), GP-B (glucose–peptone with birch sawdust), GP-A (glucose–peptone with alder sawdust) and GP-P (glucose–peptone with pine sawdust) of the *Crucibulum laeve* LE-BIN 1700.

**Figure 7 jof-11-00021-f007:**
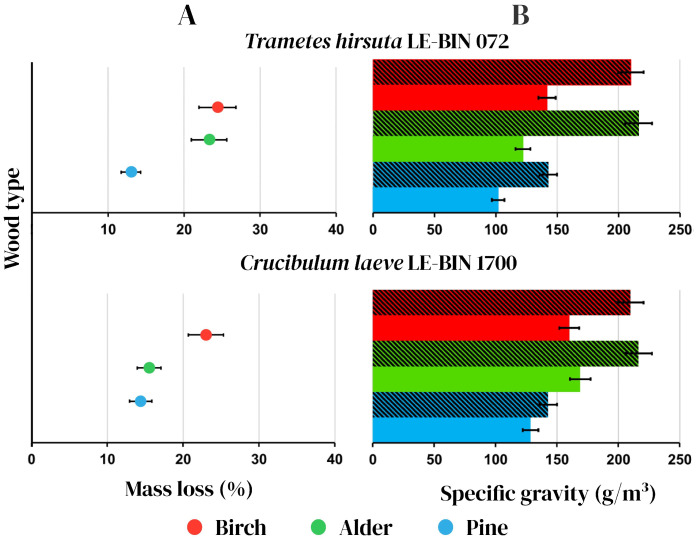
Mass loss (% original mass) (**A**) and specific gravity (**B**) of wood substrates (birch, alder and pine sawdust) caused by *Trametes hirsuta* LE-BIN 072 (**top panel**) and *Crucibulum laeve* LE-BIN 1700 (**bottom panel**) after 2 and 4 months of fungal cultivation, respectively. Shaded columns—initial specific gravity of sawdust; unshaded columns—specific gravity of sawdust after the fungal cultivation.

**Figure 8 jof-11-00021-f008:**
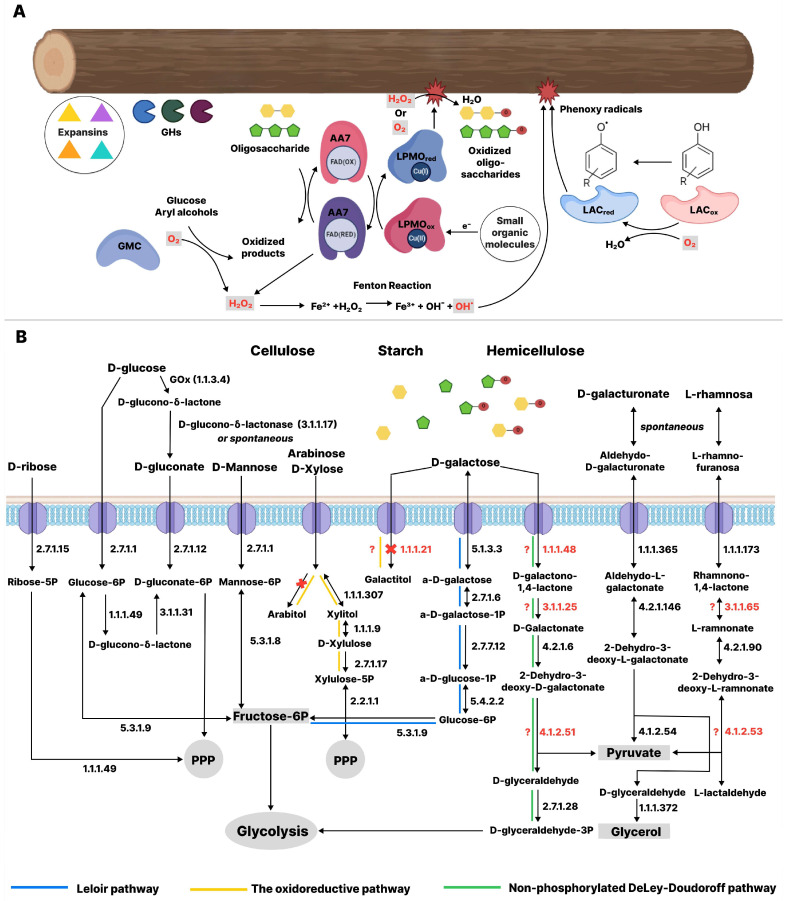
General scheme of the lignocellulose degradation enzyme machinery (**A**, **top panel**) and summary of proposed sugar catabolic pathways during lignocellulose degradation in WRF *Crucibulum laeve* (**B**, **bottom panel**). Enzymes that are absent from the *C. laeve* genome are marked in red. The question mark means that the genome of this fungus may contain analogues that are currently unknown.

**Table 1 jof-11-00021-t001:** Total esterase and lipase activities in culture liquids of *C. laeve*.

Culture Liquid	Esterase Activity, ×10^−2^ U/mL	Lipase Activity, ×10^−2^ U/mL
GP	1.0 ± 0.1	0.4 ± 0.1
GP-B	1.6 ± 0.1	0.8 ± 0.1
GP-A	2.4 ± 0.2	0.5 ± 0.1
GP-P	2.0 ± 0.1	0.4 ± 0.1

**Table 2 jof-11-00021-t002:** Phenols and reducing sugars content in culture liquids after 30 days of *C. laeve* cultivation.

Culture Liquid	Total Phenols, mg/mL		Reducing Sugars, mg/mL	
	Control	After Cultivation	Control	After Cultivation
GP	89.6 ± 0.9	48.9 ± 1.5	0.45 ± 0.05	0.36 ± 0.03
GP-B	202.3 ± 4.3	75.7 ± 0.7	0.45 ± 0.07	0.59 ± 0.06
GP-A	197.3 ± 3.2	48.8 ± 0.2	0.50 ± 0.03	0.64 ± 0.05
GP-P	129.8 ± 3.4	53.7 ± 0.5	0.41 ± 0.04	0.29 ± 0.03

## Data Availability

The original contributions presented in the study are included in the article/Appendix A, further inquiries can be directed to the corresponding author.

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
