# Peer review of "Saprotrophic Wood Decay Ability and Plant Cell Wall Degrading Enzyme System of the White Rot Fungus Crucibulum laeve: Secretome, Metabolome and Genome Investigations"

_jof, 2024, doi:10.3390/jof11010021_

Round 1

Reviewer 1 Report

This article investigates the ability of the white rot fungus Crucibulum laeve to degrade lignocellulose, providing an in-depth understanding of the process through the analysis of its secretome, metabolome, and genome. The research method is rigorous and makes a significant contribution to the field of lignocellulose biodegradation, but there are still some issues that need clarification.

1. Page 2, Line 90-93. Was the homogenization process carried out under sterile conditions? How to ensure the purity of the inoculum?

2. Page 4, Line 185-186. In the cultivation conditions, it is recommended to indicate which standard the temperature and humidity parameters refer to.

3. Page 4, Line 188. Before the wood preservation study, were the wood sawdust also dried to a constant weight under 105°C conditions? The calculation of sawdust mass loss should require consistent conditions.

4. If conditions permit, it is suggested to further study the potential molecular mechanism of C. laeve degradation of lignocellulose, including the mode of enzyme action, etc., or further statistical analysis of secretome-metabolome data (such as PCA) to reveal the interactions between different components.

5. It is suggested that the differences between C. laeve and other white rot fungi in lignocellulose degradation capacity and the potential impacts of these differences on ecosystems and biotechnology applications be further discussed in section 4. Discussion.

Author Response

Major comments

This article investigates the ability of the white rot fungus Crucibulum laeve to degrade lignocellulose, providing an in-depth understanding of the process through the analysis of its secretome, metabolome, and genome. The research method is rigorous and makes a significant contribution to the field of lignocellulose biodegradation, but there are still some issues that need clarification.

Response: The authors thank the reviewer for careful consideration of the manuscript, which allowed us to significantly improve its quality. Below are the answers point by point:

Detail comments

  1. Page 2, Line 90-93. Was the homogenization process carried out under sterile conditions? How to ensure the purity of the inoculum?

Response 1: In paragraph 2.1 of the Materials and Methods section, clarifying information has been added. Corrected to “To prepare the inoculum, the mycelial mat was homogenized with ceramic beads under aseptic conditions. For this purpose, the flasks containing the mycelium and beads were shaken at 250 rpm for 15 minutes. All the inoculations were made with on 25 mL of disrupted mycelium.” (Lines 131-134)

  1. Page 4, Line 185-186. In the cultivation conditions, it is recommended to indicate which standard the temperature and humidity parameters refer to.

Response 2: We used a modified test protocol of the EN 113-2 2021 standard to examine the abilities of two strains to degrade wood sawdust. The reference has been added in the text [Lauritz Schrader, Christoph C. Tebbe, Jochen Trautner, Christian Brischke Ability of Perenniporia meridionalis to degrade selected European-grown hardwoods // International Biodeterioration & Biodegradation 194 (2024) 105863; https://doi.org/10.1016/j.ibiod.2024.10586]. (Lines 221-228)

  1. Page 4, Line 188. Before the wood preservation study, were the wood sawdust also dried to a constant weight under 105°C conditions? The calculation of sawdust mass loss should require consistent conditions.

Response 3: Information has been added to the text. Corrected to:“The degradation capacity of the C. laeve and T. hirsuta was tested according to a modified EN 113-2 (2021) test protocol using birch, alder and pine sawdust. All wood sawdust was dried to a constant weight at (103±2) °C prior to testing. 10 g of each sawdust type was placed in 750-ml Erlenmeyer flasks to which 10 ml of tap water was added. After double autoclaving (1 atm for 30 min), T. hirsuta or C. laeve inoculum was introduced. The fungal inoculum was obtained as described in Section 2.1. The inoculated flasks were placed in an environmental chamber (ТХВ-500, NPF Termokon, Russia) and incubated in the dark at (23±2) °C and 70% relative humidity [Lauritz Schrader, Christoph C. Tebbe, Jochen Trautner, Christian Brischke Ability of Perenniporia meridionalis to degrade selected European-grown hardwoods // International Biodeterioration & Biodegradation 194 (2024) 105863; https://doi.org/10.1016/j.ibiod.2024.10586]. At the end of the incubation period (60 and 120 days for T. hirsuta and C. laeve, respectively), sawdust was removed from the flasks and dried in a hot air oven at 105 °C to constant weight. After weighing the dry sawdust before and after fungal cultivation, the percentage of mass loss due to fungal decay (ML) was calculated using the following equation: ML (%) = (m0− m0,inc) ∕ m0 × 100  (3), where m0 – the oven-dry mass before incubation [g] and m0,inc – the oven dry mass after incubation [g].

Additionally, check test flask were implemented for every type of wood sawdust setting to measure mass loss that was not caused by fungal decay (EN 113-1 2021)”. (Lines 221-237)

  1. If conditions permit, it is suggested to further study the potential molecular mechanism of C. laevedegradation of lignocellulose, including the mode of enzyme action, etc., or further statistical analysis of secretome-metabolome data (such as PCA) to reveal the interactions between different components.

Response 4: Thank you for your recommendations. In the context of the study, PCA analysis of secretome-metabolome data was not performed, since only secretome proteins were analyzed, without studying the intracellular proteome. To establish a more accurate relationship between proteins and metabolites of the fungus, it is also necessary to conduct a more in-depth and detailed study of metabolome profiles, including intracellular metabolites. This is the subject of a separate large study, which we plan to carry out and publish in the future.

  1. It is suggested that the differences between C. laeveand other white rot fungi in lignocellulose degradation capacity and the potential impacts of these differences on ecosystems and biotechnology applications be further discussed in section 4. Discussion.

Response 5: The relevant information about the enzymes of the secretomes of different fungi and their ability to destroy has been added to the section 4. Discussion. (Lines 491-528, 596-618)

Reviewer 2 Report

It should be acknowledged that this article has scientific significance. Crucibulum laeve is an important white-rot fungus that plays a crucial role in lignocellulose degradation. This study presents an extensive amount of data in an attempt to elucidate the molecular mechanisms behind Crucibulum laeve's lignocellulose degradation.

However, the organization of the article is poor, making it very difficult to read. Different sections seem to have been added merely to increase the volume of data, without clear connections or a logical progression. Instead of a coherent flow, there is a presentation of a large volume of data arranged in parallel.

For this article to be published, a systematic revision is required.

First, there is an issue of word count. The novelty and importance of this paper do not require such an extensive word count. Therefore, the manuscript must be reduced by at least one-third, retaining only the core data.

Many data could be omitted. For instance, the relationship between the proteome data and the substrate degradation comparison with Trametes species has little relevance. Similarly, the connection between the degradation of azo-labeled compounds and lignocellulose degradation should be deleted. If enzyme activity of oxidases has already been measured, why add this additional experiment? The proteomics data in this article are obtained through 2D gel electrophoresis, so the Venn diagram is unnecessary. These relationships are already visible from the proteomics profile, which should label protein spots with corresponding protein number (1, 2, 3,)

Figure 5 already lists all identified proteins, so there is no need to devote entire paragraphs to describing each detected protein individually. This detailed description suggests a lack of clarity regarding which protein is the most significant.

Similarly, why does the conclusion section match the length of the discussion? The experimental data is not enough to support such big conclusion.

It is essential to remove all irrelevant content and focus on a central theme. Within this theme, different experimental results should support and build upon one another, demonstrating a clear, progressive relationship, rather than simply showcasing disparate data.

none

Author Response

It should be acknowledged that this article has scientific significance. Crucibulum laeve is an important white-rot fungus that plays a crucial role in lignocellulose degradation. This study presents an extensive amount of data in an attempt to elucidate the molecular mechanisms behind Crucibulum laeve's lignocellulose degradation.

Response: The authors thank the respected reviewer for his good assessment of the significance of this study and valuable comments

However, the organization of the article is poor, making it very difficult to read. Different sections seem to have been added merely to increase the volume of data, without clear connections or a logical progression. Instead of a coherent flow, there is a presentation of a large volume of data arranged in parallel.

For this article to be published, a systematic revision is required.

Response: We then carefully considered all the reviewer's comments that specified this general position. In doing so, we also took into account the recommendations of two other reviewers. As a result, the new submitted version of the article has been substantially changed and restructured. Below, we provide comments on the recommendations of the esteemed reviewer, relating to specific sections.

First, there is an issue of word count. The novelty and importance of this paper do not require such an extensive word count. Therefore, the manuscript must be reduced by at least one-third, retaining only the core data.

Response: We reviewed the materials taking into account this reviewer's recommendation and restructured the text, removing several additional experiments and shortening the manuscript. Note that the study included a consistent description of: (1) the ability of the fungus Crucibulum laeve to grow and degrade wood sawdust; (2) identification of exosecretome proteins involved in the process of sawdust biodegradation; (3) identification of metabolites, including sugars, formed during sawdust biodegradation; (4) assessment of the fungus' ability to metabolize sugars formed during sawdust saccharification - an in silico analysis of carbohydrate metabolic pathways was performed. Therefore, the description of all these stages should be retained in the article to present to the readers the relationships between the ability to grow and degrade sawdust and the enzymes involved in this process in the fungus Crucibulum laeve. Accordingly, the reduction in the number of SLAVs in the core text is less than the 30% recommended by the reviewer.

Many data could be omitted. For instance, the relationship between the proteome data and the substrate degradation comparison with Trametes species has little relevance. Similarly, the connection between the degradation of azo-labeled compounds and lignocellulose degradation should be deleted. If enzyme activity of oxidases has already been measured, why add this additional experiment? The proteomics data in this article are obtained through 2D gel electrophoresis, so the Venn diagram is unnecessary. These relationships are already visible from the proteomics profile, which should label protein spots with corresponding protein number (1, 2, 3,)

Response: The data concerning “the degradation of azo-labeled compounds” and the Venn diagram were removed from the article. As for “the relationship between the proteome data and the substrate degradation comparison with Trametes species has little relevance”, we disagree with the reviewer here. Since representatives of the genus Trametes are reference WRF in terms of lignocellulose biodegradation and their lignocellulolytic enzymatic complex is well studied. In this regard, in our work we compared the sawdust degradation capacity of two representatives of the WRF orders Polyporales (Trametes hirsuta) and Agaricales (Crucibulum laeve). And also in the Discussion section we compared the data obtained in this work and the literature data concerning the exoproteomic profiles of representatives of the WRF orders Polyporales and Agaricales.

Figure 5 already lists all identified proteins, so there is no need to devote entire paragraphs to describing each detected protein individually. This detailed description suggests a lack of clarity regarding which protein is the most significant.

Response: We agree that the commentary to Figure 5 should not be transformed into a description of the properties of all the proteins shown there. Our text examines the key enzymes involved in the breakdown of lignocellulose and demonstrates processes that are also examined in other sections of the article to confirm the connections of these processes with proteomic data. Therefore, when revising this section, we only simplified and somewhat shortened the descriptions provided.

Similarly, why does the conclusion section match the length of the discussion? The experimental data is not enough to support such big conclusion.

Response: We have substantially shortened the conclusion by removing a number of general comments and using some specific considerations to strengthen the discussion of the results.

It is essential to remove all irrelevant content and focus on a central theme. Within this theme, different experimental results should support and build upon one another, demonstrating a clear, progressive relationship, rather than simply showcasing disparate data.

Response: We also note that, taking into account the recommendations of other reviewers, a number of focused comments on certain experiments have been added to the Discussion section. The Discussion section has also been divided into subsections. We hope that this will improve the clarity of the data presentation and eliminate the shortcomings noted in this review.

Reviewer 3 Report

Add more information to the intro regarding laccases, aryl-alcohol oxidases, lytic polysaccharide monooxygenases, and non-specific peroxygenases - modes of action, typical producers, what these enzymes breakdown, etc. 

Have other isolates of C. laeve been characterized? Why this specific isolate? 

What were the concentrations of the various separate lignocellulosic components used in culture growth rate studies?

Were the decay tests used following a specific standard? 

Was oxalate decarboxylase found in the genome? 

Line 72: consider changing first attempt to study to first attempt to characterize

What wood species was C. laeve isolated from? - please add to the text

Line 203 should read pooled together not pulled together

Line 513 should read wood xylotroph not wound xylotroph

Figure 9: would help if the fungal organisms were added to the graph

Line 537: States you evaluated C. laeve decay living on forest liter - what do you mean by this? You did not evaluate C. laeve decay of forest litter just decay of alder, birch and pine. 

Lines 589-595 can you expand on this a bit, specifically why such big differences in enzymes between these 2 WRF? 

Author Response

Major comments

(1) Add more information to the intro regarding laccases, aryl-alcohol oxidases, lytic polysaccharide monooxygenases, and non-specific peroxygenases - modes of action, typical producers, what these enzymes breakdown, etc. 

Response 1: The authors are grateful to the reviewer for valuable comments, which allowed us to significantly improve the manuscript. Information about the enzymes mentioned was added to the text. (Lines 57-84)

(2) Have other isolates of C. laeve been characterized? Why this specific isolate? 

Response 2: Information has been added to the text in the introduction section. (Lines 96-109):

To the best of our knowledge, this study represents the first attempt to characterize the saprotrophic wood-degrading abilities of a WRF C. laeve LE-BIN 1700 (Agaricales, Nidulariaceae) from the Komarov Botanical Institute Basidiomycetes Culture Collection (LE-BIN, St. Petersburg, Russia) isolated from twig (Western Caucasus, Adygei Republic, Russia). We have previously shown in biochemical plate tests that this C. laeve isolate differs from other tested white rot fungi in its enzymatic activity spectrum [Moiseenko, K.V.; Glazunova, O.A.; Shakhova, N.V.; Savinova, O.S.; Vasina, D.V.; Tyazhelova, T.V.; Psurtseva, N.V.; Fedorova, T.V. Fungal Adaptation to the Advanced Stages of Wood Decomposition: Insights from the Steccherinum ochraceum. Microorganisms 2019, 7, 527. https://doi.org/10.3390/microorganisms7110527]. In these tests, two substrates (ABTS and Azur B) were used to assess the overall oxidative capacity of the fungal mycelium, and carboxymethyl cellulose (CMC) was used as a substrate to assess the overall cellulolytic activity. While ABTS is considered to be an easily degradable substrate that can potentially be oxidized by Lacs and PODs, Azur B can only be degraded by enzymes with high redox potential, such as ligninolytic peroxidases. The C. laeve isolate LE-BIN 1700 demonstrated a high capacity to degrade ABTS, while the capacity to degrade Azur B and CMC was not detectable.

(3) What were the concentrations of the various separate lignocellulosic components used in culture growth rate studies?

Response 3: The concentration of the various separate lignocellulosic components used in culture growth rate studies was 1 g/L (see Section of “Materials and Methods”, paragraph 2.2; (Lines 147-150): «Carboxymethyl cellulose (CMC, Sigma, St. Louis, MO, USA); birch and larch xylan 106 (Sigma, St. Louis, MO, USA); lignosulfonate (Sigma, St. Louis, MO, USA); starch (MP 107 Biomedicals, France) and pectin (Shanghai Acmec Biochemical, China) were used as separate components of lignocellulose (1 g/L).»

(4) Were the decay tests used following a specific standard? 

Response 4: Decay tests were conducted as described in a number of studies. We used a modified test protocol of the EN 113-2 2021 standard to examine the abilities of two strains to degrade wood sawdust. The corresponding references have been added to section 2.7. in the Materials and Methods. [Lauritz Schrader, Christoph C. Tebbe, Jochen Trautner, Christian Brischke Ability of Perenniporia meridionalis to degrade selected European-grown hardwoods // International Biodeterioration & Biodegradation 194 (2024) 105863; https://doi.org/10.1016/j.ibiod.2024.10586]. (Lines 221-237)

(5) Was oxalate decarboxylase found in the genome? 

Response 5: Indeed, the protein Bicupin, oxalate decarboxylase/oxidase, was found in the genome. Two different activities are known for members of this family: oxalate decarboxylase (EC 4.1.1.2) and oxalate oxidase (EC 1.2.3.4).  Although the latter activity has more often been found in distantly related monocupin (germin) proteins. Added to text (Lines 678-685).

 (6) What wood species was C. laeve isolated from? - please add to the text

Response 6: In paragraph 2.1 of the Materials and Methods section information has been added (Lines 116-125). Corrected to “The fungal strain of Crucibulum laeve (Hudson, 1802) Kambly, 1936 was isolated (September 17, 2003) from basidiospores collected from a fallen dead branch in the fir-beech forest (Western Caucasus, Adygei Republic, Kavkazsky nature reserve, vic. Guzeripl reserve station, Russia; N 43°59′; E 40°07′). After morphological and genetic verifications, the strain was deposited in the Komarov Botanical Institute Basidiomycetes Culture Collection (LE-BIN; St. Petersburg, Russia) as C. laeve LE-BIN 1700. The sequence of its ITS1-5.8S rRNA-ITS2 region is available at the NCBI GenBank accession MK795850. The fungal strain Trametes hirsuta LE-BIN 072 was also obtained from LE-BIN (GenBank accession number: MK795848) [Moiseenko, K.V.; Glazunova, O.A.; Shakhova, N.V.; Savinova, O.S.; Vasina, D.V.; Tyazhelova, T.V.; Psurtseva, N.V.; Fedorova, T.V. Fungal Adaptation to the Advanced Stages of Wood Decomposition: Insights from the Steccherinum ochraceum. Microorganisms 2019, 7, 527. https://doi.org/10.3390/microorganisms7110527].”

Detail comments

Line 72: consider changing first attempt to study to first attempt to characterize

Response: Corrected to: « To the best of our knowledge, this study represents the first attempt to characterize the saprotrophic wood-degrading abilities of a WRF C. laeve LE-BIN 1700 (Agaricales, Nidulariaceae) from the Komarov Botanical Institute Basidiomycetes Culture Collection (LE-BIN, St. Petersburg, Russia) isolated from twig (Western Caucasus, Adygei Republic, Russia)» (Lines 96-100).

Line 203 should read pooled together not pulled together

Response: Corrected to: “For the analysis of exoproteomes and metabolomes, cultural liquids from biological replicates were pooled together” (Lines 248-249)

Line 513 should read wood xylotroph not wound xylotroph

Response: Corrected to: “the primary wood xylotroph (T. hirsuta) degrades hardwood sawdust more potently than coniferous one” (Lines 407, 418).

Figure 9: would help if the fungal organisms were added to the graph

Response: Fungal organisms have been added to the figure. The numbering of the figure has been changed and now it is Fig.7

Line 537: States you evaluated C. laeve decay living on forest litter - what do you mean by this? You did not evaluate C. laeve decay of forest litter just decay of alder, birch and pine. 

Response: We meant that we were assessing the destruction by the fungus C. leave, which naturally lives on forest litter. The phrase in the text was corrected to: “In this paper, we evaluated the efficiency of the WRF C. laeve (Agaricales, Nidulariaceae) in destroying birch, alder and pine sawdust, and also analyzed the secretomic and metabolic profiles resulting from the fungal cultivation on the sawdust-containing media”. (Lines 441-444)

Lines 589-595 can you expand on this a bit, specifically why such big differences in enzymes between these 2 WRF? 

Response: The relevant information has been added to the section 4. Discussion (Lines 491-528, 596-618).

Round 2

Reviewer 1 Report

All issues have been revised.

The manuscript can be published as it stands.

Author Response

Major comments: All issues have been revised.

Response: The authors thank the respected reviewer for his valuable comments, according to which the manuscript has been significantly improved.

Detail comments: The manuscript can be published as it stands.

Response: The authors would like to thank the reviewer for the favourable evaluation of the manuscript